# Spatial Metadata Usability Evaluation

**Mohsen Kalantari [1,\*], Syahrudin Syahrudin [2] , Abbas Rajabifard [1], Hardi Subagyo [3] and Hannah Hubbard [1]**

1. Department of Infrastructure Engineering, The University of Melbourne, Victoria 3010, Australia; abbas.r@unimelb.edu.au (A.R.); hubbardh@unimelb.edu.au (H.H.)
2. Indonesian Geospatial Agency, Jakarta 16911, Indonesia; syahrudin@big.go.id
3. Independent Researcher, Bogor 16115, Indonesia; hardi.subagyo@gmail.com
* Correspondence: mohsen.kalantari@unimelb.edu.au

**Abstract:** Spatial metadata is a critical part of any spatial data infrastructure, which enables the organising, sharing, discovery and use of spatial data. This paper highlights a knowledge gap in the usability of the metadata systems for the end–users. It then addresses the gap by applying the User Centred Design approach to investigate the usability of metadata records. The research engages with end–users concerning efficiency and effectiveness of metadata systems, and end–users' satisfaction and expectations. The results indicate significant gaps with the effectiveness and efficiency of metadata systems for spatial data discovery and selection. Inconsistency and irrelevant information in the metadata records were found in the title, keywords, abstracts, data quality and other elements of the metadata. Additionally, essential improvements were identified for user interfaces. Discouraging presentation of the metadata is a prominent problem found in the interface of the metadata systems.

**Keywords:** spatial data infrastructure; spatial metadata; geoportal; user centred design; usability; Australia

## 1. Introduction

Spatial metadata plays an important role in promoting spatial data sharing and re–use and supporting local and global development initiatives that require spatial data to manage, monitor and measure the development. It contains information about geographic or spatial dataset descriptions, e.g., contents, structure, quality, and reference system that will help spatial data users to discover and determine the suitability of the data for their purposes through networked spatial data catalogue systems, e.g., Spatial Data Infrastructure (SDI) and geo–portals.

However, the usability of spatial metadata for data users is questioned, since the metadata is created following standards originally designed for spatial data producers for data inventory purposes. Moreover, with the limitless possible applications of the metadata, at the very least we need to know how usable the metadata for spatial data users is for discovering and selecting appropriate spatial data. With the growing number of spatial data users outside the spatial field with specific knowledge, the requirements and expectations of spatial data users should be considered.

Spatial metadata usability is a combination of the usability of the metadata records and usability of the user interface. The state of the research in spatial metadata is limited to a few broad areas. Some investigations are dedicated to creating complete and consistent metadata records [1–9]. These papers reveal that most challenges for automatic metadata creation and updating relate to the type of information included in spatial data characteristics. Only a few pieces of information about spatial data can be automatically extracted and stored into metadata, e.g., the reference system, projection, date of last update, and lineage statement. All other information (e.g., abstracts and data quality descriptions) still rely on manual input by humans. There are also attempts to put emphasis on

the involvement of users in metadata creation. There are also quite a number of studies that improve the discovery of spatial data using semantics and ontologies [10–22]. The papers show that there have been attempts to involve users in the creation of metadata, both direct and indirect. For example, direct involvement has been conducted by directly engaging users to provide information for metadata creation. On the other hand, indirect or implicit user involvement is conducted by collecting and analysing users' search words and assigning the results as tags for the spatial data.

To address the current knowledge gaps in spatial metadata usability, this paper reports the results of an experiment for investigating the usability of spatial metadata by employing two User Centred Design methods, namely Think–Aloud Protocol usability testing (TAP) and semi–structured interviews. TAP records the verbal expressions of participants' thoughts during the process of task completions and researcher's notes taken while observing the process. The semi–structured interviews are used to reveal detailed information of participants' experiences by encouraging them to recall specific incidents during the task completions. Both data captured by TAP and semi–structured interviews are then analysed and extracted to determine the usability of the spatial metadata, identify the usability problems, describe the user experience and reveal the user expectations.

Section 2 starts with an explanation of the data analysis method followed by experiment design, the reasoning for material and participants selection. Section 3 of the paper provides a summary of the results. Sections 4 and 5 analyse the outcomes of the data discovery and data selection task by participants. Section 6 collects insights from the participants' opinions during the experiment. Section 7 provides a discussion on the findings of the paper, and Section 8 concludes the research.

## 2. Method

### 2.1. Data Analysis Methods

The method used in the analysis process is adapted from Protocol Analysis (PA) and content analysis. PA is capable of producing deep and accurate analysis from TAP data, as these methodologies are qualitative in nature and designed to obtain rich and deep data from small quantities of samples [10–12]. PA consists of several progressive steps as shown in Figure 1.

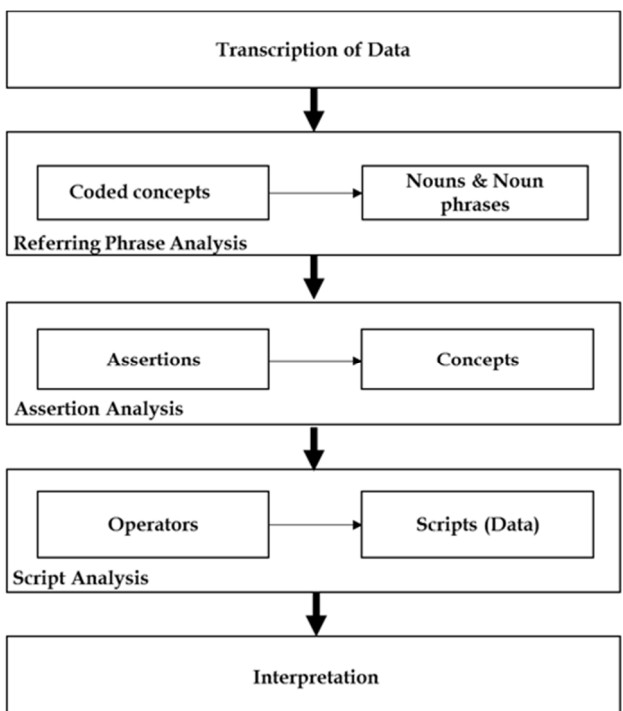

**Figure 1.** Protocol analysis framework [2].

The analysis process begins with Transcription of Data (ToD), where recordings and researcher's notes during the testing are transformed into textual transcripts. The next step is Referring Phrase Analysis (RPA). It is done by highlighting all the nouns and noun phrases identified in each participants' verbal data and attaching codes to those phrases in accordance with code references within the spatial data search activities. Based on the provisional coding results of the RPA, the investigator then groups each code according to each relevant action during the spatial data discovery and selection. The next step is Assertions Analysis where the investigator identifies the set of assertions made by subjects to determine how relationships were being formed between concepts during problem solving. As part of the analysis, the purposes of subjects' assertions are identified. The last step is Script Analysis to identify and provide an overall description of the reasoning process during problem solving and allowing the investigator to illustrate the problems identified by the subjects during the problem–solving process.

## 2.2. Experiment Design

The objective of this experiment is to collect and analyse data to answer research questions based on a selected usability evaluation method as illustrated in Figure 2. The experiment was designed based on the usability evaluation framework, TAP. The TAP data collection method utilises semi–structured interviews and post–questionnaires, and spatial metadata usability is evaluated using three criteria: usefulness, effectiveness and user satisfaction (middle blocks in Figure 2). The design was based on a task–oriented activity where participants were given a set of tasks concerned with spatial metadata, following a scenario to achieve specified goals (top blocks in Figure 2). During the completion of the tasks, they were asked to actively and continuously express their thoughts and opinions verbally, as well as answering questions given by the researcher (top blocks in Figure 2). At the end of the experiment, the participants answered some questions in semi–structured interviews to provide the researcher with detailed and deeper information related to the given tasks and based on their experiences during the TAP (top blocks in Figure 2).

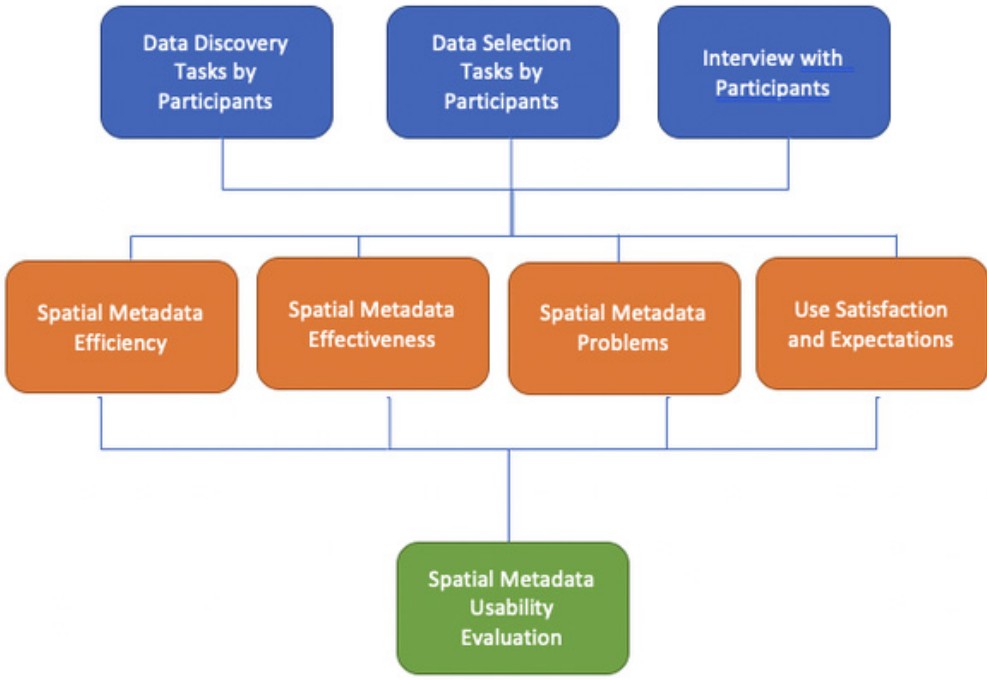

**Figure 2.** Diagrammatic view of experiment design.

## 2.3. Participants

For this study, five spatial data users from different educational backgrounds were selected; civil engineering, spatial science and computer science. They were working on disaster management.

The number is optimal for usability testing method using TA protocol, since it is able to identify the most usability problems with optimum benefit–cost ratio [12]. In addition to that, the study is a qualitative and exploratory in nature, focusing on the extraction of user's insight about spatial metadata. This type of study usually needs a small number of participants with a long time of participant–researcher interaction, as opposed to a quantitative study which focuses on statistics, using a large number of participants and little if any participant–researcher interaction [10].

### 2.4. Materials

Spatial metadata records were selected from two data directories or portals: Victoria Data Directory (VDD) and Australian Spatial Data Directory (FIND). The portals were selected based on the following criteria. The metadata is (1) created in accordance with international metadata standards (ISO 19115); (2) available publicly and accessible via the internet; (3) the VDD represents open data platform and VDD represents spatial data catalogue. VDD is a state level portal that connects and shares spatial metadata repositories with official institutions in Victoria State, while FIND is a national level portal consisting of metadata nodes from national institutions such as Geoscience Australia and Bureau of Meteorology.

Other materials were a set of questions designed as guide questions for semi–structured interviews and written instructions for the participants containing scenarios, tasks and objectives. Another important technique for this study was the use of audio recording for collecting participants' verbal expressions and opinions during the completion of the test.

### 2.5. Scenario

For this experiment, the scenario was to develop an evacuation map for Greater Melbourne and the East Coast of Australia (Queensland and New South Wales). The set of spatial data required for this activity follows:

- Elevation dataset with minimum scale of 1:25,000
- Hydrographic datasets (e.g., rivers, dams, ponds, etc.) with minimum scale of 1:25,000
- Meteorology datasets of information (e.g., rainfalls)
- Road network or transportation network datasets with minimum scale 1:10,000 for urban area and 1:100,000 for rural area
- Soil type maps with minimum scale of 1:100,000

To acquire the appropriate data, participants searched the Victorian Data Directory (VDD) and Australian Spatial Data Directory (FIND) websites.

Tasks—following the given scenario, the participants were allocated three main tasks, to:

1. find the required spatial data from discovery systems (VDD and FIND);
2. determine the suitability of spatial data for the given scenario; and
3. determine how to obtain and appropriately use the spatial data.

The first task was designed to investigate the usability of metadata, as well as discovery systems, in helping users to find and locate the required spatial data. The second was for participants to use the information from metadata records to assess the suitability of the spatial data for the given purpose. The third question targeted the utility of spatial data for the given application. They were asked to verbally express their thoughts about the tasks and spatial metadata in detail.

### 2.6. Procedure

The usability test was conducted in a prepared laboratory environment using the following procedures:

1.  Orientation Session—each participant attended orientation prior to performing data discovery and selection tasks. During the orientation, the participants were given an explanation about the purpose of the study, the scope and expectations of the experiment and the tasks to be performed. Written instructions were also given. In this session, participants were allowed to ask questions related to the study and the tasks, and at the end of this session participants were asked to sign a consent form. This session lasted 5 min.

2.  Task performance—each participant performed and completed three tasks as described in the task section.

    *   The first task was searching and locating the spatial metadata required for the given scenario from the spatial data directory. Participants were given 30 min to search and find the required spatial data from the discovery systems. They were asked to stop the task when the time was finished. At that stage, success rate was observed.
    *   The second task was to decide whether the data was suitable for the given project or not, by examining the spatial metadata records obtained from the first task or, if the participant had been unable to find relevant records, the pre–selected metadata records prepared by the researcher.

3.  Interviews—after completing the tasks, participants were asked to fill out a questionnaire and to explain their answers to the researcher. The questions were designed to get participant's opinions about their experiences working with the spatial metadata during the data discovery and selection process.

During the experiment, the participants were accompanied by a representative from the research team. The representative's role was to give instructions and explanations about the test during the orientation session, answer questions raised by the participants during the performance of the tasks, and to make sure that the participants stayed focused on the objectives. The representative also asked participants the prepared questions when an interview stalled or to elicit more detailed insights from participants about the metadata.

## 3. Results

Two data sources were collected from the experiment: audio recordings and semi–structured interviews based on a set of questions. As an exploratory study, analysis methods used were mainly qualitative. TAP analysis was used to analyse the TAP data. The interview data was analysed using content analysis.

*Think–Aloud Protocol (TAP)*

During the spatial data discovery and selection process, participants' verbal expressions were recorded. Each record was then transcribed and analysed separately using the protocol analysis. The transcriptions were analysed by identifying referring phrases (RPA) and each phrase was given codes based on the reference codes listed in Table 1. Examples of the phrases, and the codes which resulted from this process, are presented in Table 2.

**Table 1.** Reference codes used in Protocol Analysis [2].

| Concept | Definition |
| --- | --- |
| Action | The manner or method or performing; a thing done |
| Element | A piece of information describing certain characteristics of spatial data |
| Sign | Objective information indicative of status of the information or tools |
| Time | A chronological reference |
| Needs | Information, tools or actions required as to discover or select data |
| Value | A rating of usefulness, importance or worth |
| Goal | A specific goal for an action |

**Table 2.** Example of Referring Phrase Analysis (RPA) results from TAP for spatial metadata usability testing.

| Phrases | Coded Concept(s) |
|---|---|
| ✓ We need a topographic or elevation map of Melbourne | Data |
| ✓ (I did the search for topography, so) I see some results | Action, Element, Signs |
| ✓ If I do not find the results, I can search other terminology | Sign, Action, Element |
| ✓ (and I see) eight pages of the results | Signs |
| ✓ Why can't it be sorted based on the most relevant ones | Signs, value |
| ✓ The title doesn't mention the scale | Element, Signs, Needs |
| ✓ We need a scale here | Needs |
| ✓ The scale is not mentioned here | Needs, Signs |
| ✓ It could be good to put the scale (in the title) | Value, Needs |
| ✓ (If we use) elevation map, 37 datasets were found | Action, Element, Signs |
| ✓ (By applying) the most relevant, it gives me 1300 | Action, Signs |
| ✓ I tried to put the newest back, and it gives me 2000 results | Action, Signs |
| ✓ This might be related to technical issues of the system | Signs, Needs |
| ✓ We go with the newest first, but we cannot go through around 3000 results | Action, Signs |
| ✓ I think users will look at the title on the results | Action, Element |
| ✓ the title is not adjusted very well | Element, Signs |
| ✓ and when I tried to search using region (as a keyword) I cannot find exact data for Melbourne | Action, element, Signs |
| ✓ If I want to find a specific region in Melbourne, like Southbank, I cannot find it easily | Needs, Element, Signs |
| ✓ The system does not provide any advanced search | Needs, Signs, Needs |
| ✓ It does not have any filters | Signs, Needs |
| ✓ But it has license, wms, shapefile output, vegetation, etc. | Signs, Element |
| ✓ So maybe it is better to try these tags (to see what is going to happen) | Needs |
| ✓ Let's see this HY waterpoint, I don't know what it stands for, maybe hydro | Element, Sign, Element |
| ✓ I think the filters (for the search) are not sufficient and the data are not well tagged | Needs, Sign, Data, Sign |
| ✓ because it does not have enough tags and the categories are lacking | Sign, Needs, Needs, Sign |
| ✓ I do not know how to filter the scale | Sign, Action, Element |
| ✓ No tools to choose the scale | Sign, Action, Element |
| ✓ Some of the titles have scale in them, and some don't | Element, Sign, Needs, Sign |
| ✓ We can spend a lot of time searching for each data | Sign, Action |
| ✓ We can use different keywords, and maybe after half an hour or more, we can find the one you are looking for | Action, Element, Time, Sign |
| ✓ But the efficiency of the search tools and the database should be higher than that | Needs, Needs, Value |
| ✓ I need to find the (potential) data in one minute, otherwise, I could spend a lot of time | Action, Data, Time, Sign |

From Table 2, we can see phrases and respective concepts (codes) resulted from the participants' verbal expressions during the spatial data discovery and selection. These phrases and codes identify all the vocabulary and concepts the participants' focused on during the process.

The phrases and the codes were then further analysed to determine the assertions of each phrase. These assertions were then used to understand how the relationships were being formed during the spatial data discovery and selection. Table 3 shows the phrases and their respective assertions.

The final stage of the analysis process was conducted by implementing script analysis, which assigns a set of operators to the phrases to identify the process of reasoning during spatial data discovery and selection. Three operators were used: suggest, study and conclude. These operators represent the interactions in the reasoning process of spatial data discovery and selection. Table 4 provides an explanation of the operators.

**Table 3.** Example of assertions analysis of the TAP data.

| | Data | Assertion |
|---|---|---|
| ✓ | I do not know how to filter the scale | Causal [3] |
| ✓ | No tools to choose the scale | Causal |
| ✓ | Some of the titles have scale in them, and some do not | Connotative [2] |
| ✓ | We can spend a lot of time searching for each data | Causal |
| ✓ | We can use different keywords | Indicative [1] |
| ✓ | And maybe after half an hour or more, we can find the one you look for | Causal |
| ✓ | But the efficiency of the search tools and the database should be higher than that | Connotative |
| ✓ | I have to be able to find the (potential) data in one minute, otherwise, I could spend a lot of time | Causal |

[1] Indicative assertions form relationships of significance of an action. [2] Connotative assertions form relationships of meaning or state. [3] Causal assertions for relationships of cause and effect.

**Table 4.** Example of script analysis of the TAP data.

| | Data | Operator |
|---|---|---|
| ✓ | we need a scale here | Suggest [1] |
| ✓ | The scale is not mentioned here | Study [2] |
| ✓ | It could be good to put the scale (in the title) | Suggest |
| ✓ | (If we use) elevation map, 37 datasets were found | Study |
| ✓ | (By applying) the most relevant, it gives me 1300 | Study |
| ✓ | I tried to put the newest back, it gives me 2000 results | Study |
| ✓ | this might be related to technical issues of the system | Conclude [3] |
| ✓ | I think users will look at the title on the results | Suggest |
| ✓ | the title is not adjusted very well | Study |
| ✓ | and when I tried to search using region (as a keyword) I cannot find exact data for Melbourne | Conclude |
| ✓ | If I want to find a specific region in Melbourne, like Southbank, I cannot find it easily | Conclude |
| ✓ | The system does not provide any advanced search | Study |
| ✓ | It does not have the filters | Study |
| ✓ | I could use (filters) to filter the data | Suggest |

[1] Suggest means to consider an action or information to achieve a specific goal. [2] Study means to observe or consider information or facts carefully. [3] Conclude means to decide on the significance, value or results of an action.

From the tables presented above, we can identify the participants' activities, goals and expectations, as well as relationships between them and their reasoning when they try to discover and select spatial data using the metadata via the user interfaces.

The results of script analysis were then used as the basis for identifying the effectiveness, efficiency and user satisfaction, as well as identifying usability problems and user expectations for spatial data discovery and spatial data selection.

## 4. Analysing the Outcomes of Spatial Data Discovery Tasks

In spatial data discovery users use the user interface to interact with metadata to discover the data for the given purpose according to the scenario. Effectiveness, efficiency, user satisfaction, problems and user expectations were then identified by interpreting the results.

### 4.1. Effectiveness

Effectiveness of the spatial metadata is a combination between the discovery systems or searching tools and the metadata records. To this end, effectiveness was measured by the successfulness of the participants in finding and discovering potential spatial data for the given scenario from the two discovery systems: VDD and FIND. From the script analysis results, we can identify the successful rate of the discovery by looking at the conclude operator, which is related to significance, value or results of an action.

To acquire a comprehensive result, conclude phrases were examined and grouped based on the tones of the phrases; positive, neutral and negative tones. Positive is when a phrase indicates a successful result and negative if it indicates otherwise. Neutral is a tone where a phrase is neither positive nor negative.

"I see some results"—Positive conclude

"I cannot find exact data for Melbourne"—Negative conclude

"can we assume that we are failing to discover the map?"—Neutral conclude

By only analysing the conclude phrases throughout the analysis document, it is hard to make a clear statement about the effectiveness of the spatial metadata for data discovery. Nevertheless, the concludes provide indications of ineffectiveness. The example concludes phrases given above are a participant's verbalisations during the discovery process of the Melbourne Topographic Map. Meteorology data provides other examples of conclude phrases:

"so, the meteorological did not respond that much"

"if you look for the rain, you only get one result"

Most of the conclude phrases do not provide a clear indication that the data discovery process can be successfully completed or not. In the examples above, the participant tried to finish the searching with a doubtful decision. It is interesting to investigate why this participant came to that decision. To find out, we look at other phrases, study and suggest, which, together with conclude, form the whole process of reasoning, as illustrated in Table 5.

**Table 5.** Script analysis data for topographic data.

|  | Data | Operator |
|---|---|---|
| ✓ | We need a scale here | Suggest |
| ✓ | The scale is not mentioned here | Study |
| ✓ | It could be good to put the scale (in the title) | Suggest |
| ✓ | (If we use) elevation map, 37 datasets were found | Study |
| ✓ | (By applying) the most relevant, it gives me 1300 | Study |
| ✓ | It tried to put the newest back, and it gave me 200 results | Study |
| ✓ | I cannot find the exact data for Melbourne | Conclude |
| ✓ | Some of the titles have scale in them, and some do not | Study |
| ✓ | We can spend a lot of time searching for each data | Conclude |

Looking at the whole reasoning process, the participants have negative concludes because they could not find the scale in the title. One participant faced 37 initial results, 1300 results after clicking the most relevant and 2000 after another click on the newest results. This participant found that the results, represented in a list of titles, should have included a scale. A scale would have greatly assisted them to locate the required spatial data, as the participant did not want to spend a lot of time searching titles by clicking on them one by one.

Another participant showed a different attitude when facing the same situation, as they just clicked on a title that they thought was the most relevant to the required data.

"There are some problems with this. I type elevation, and this one is the most related, but it does not come first"

"I think the accuracy is enough, I think if we have this, the accuracy will be enough"

"so, can I look for another one?"

The participant ended up spending a lot of time, more than the time given to do the search, by clicking and opening each and every title in the result list to find the relevant data for the given scenario.

There were recurring groups of phrases, positive, negative and neutral, from the data during the spatial data discovery process, which indicate that spatial metadata effectiveness is subject to the quality of the metadata. The quality of the metadata is determined by the following: completeness, consistency and relevance of the information presented to users; the intuitiveness of the user interface against the user needs; detailed criteria for data search filters; and willingness of users to do further detailed search within the results.

The results indicate that spatial metadata, along with the searching tool, can be used to find and discover spatial data, but users found some obstacles that hampered them from performing the task efficiently, as explained in the next section. These obstacles might prevent users from reaching the task goal, as they tended to stop the searching right away when they found that the search was not going as they expected. They would try other ways to find the data, such as finding information about the data from an internet search engine and contacting the producer directly.

## 4.2. Efficiency

Spatial data users can search the required potential data from the websites, but some effort and time was necessary to find and discover the data. The following concludes indicate that the data discovery was not a straightforward process, where participants could not get the expected results in the given time:

"We can spend a lot of time searching for each data"

"and maybe after half an hour or more, we can find the one you look for"

"But the efficiency of the search tools and the database should be higher than that"

"I have to be able to find the (potential) data in one minute, otherwise, I could spend a lot of time"

During the process, similar concludes were recurring to the point that most of the users did not want to continue their current data searches and asked to move to another search.

"If I cannot find it in like one minute, it means I have to move on to the next search"

The researcher's notes during the process of this task also indicate that none of the participants could discover all the required data in the given time.

These results indicate that the spatial metadata is able to help users to discover spatial data from the websites, but there are efforts required to increase the efficiency of the process. What needs to be done to enhance efficiency may be determined by first revealing the problems and user expectations, as explained in the next sections, and addressing the problems by enhancing the metadata and the user interface simultaneously.

## 4.3. Usability Problems

Participants faced several problems in their attempts to discover the required spatial data using the websites. The problems are related to both the website (user interface) and the metadata (content). These problems can be identified by examining the results in the following examples. Figure 3 illustrates some of the problems identified during spatial data discovery as mentioned by the participants, e.g., unmatched keywords and results, tags/filters that are not easily recognised nor sufficient to participants' needs.

"No tools to choose the scale"

"Some of the titles have scale in them, and some do not"

"Why can we not short it based on the most relevant ones"

"The title does not mention the scale"

"(By applying) the most relevant, it gives me 1300. I tried to put the newest back, and it gives me 2000 results"

"and when I tried to search using region (as a keyword) I cannot find exact data for Melbourne"

"The system does not provide any advance search"

"I think the filters (for the search) are not enough and the data are not well tagged"

"Interesting part is the titles are not very accurate, I think they come from the title of database or datasets"

"This tag is helpful, otherwise you see the title might not be relevant"

"you have to have some sort of scale, so you can filter."

"One (thing that) I could do is open this one to find out the scale because it is not mentioned here. Look how many data to open to see the scale."

"if you are using this system after a while you find that tags are not helpful"

"what is this VIF 2015? I do not know, I have to open it. I have to see the data to understand VIF, and it takes time."

"That's why it is important to put the name, not the abbreviations."

"the title is about elevation, but here the search is land."

"when I typed the keywords and the same come out is totally different from what I expected"

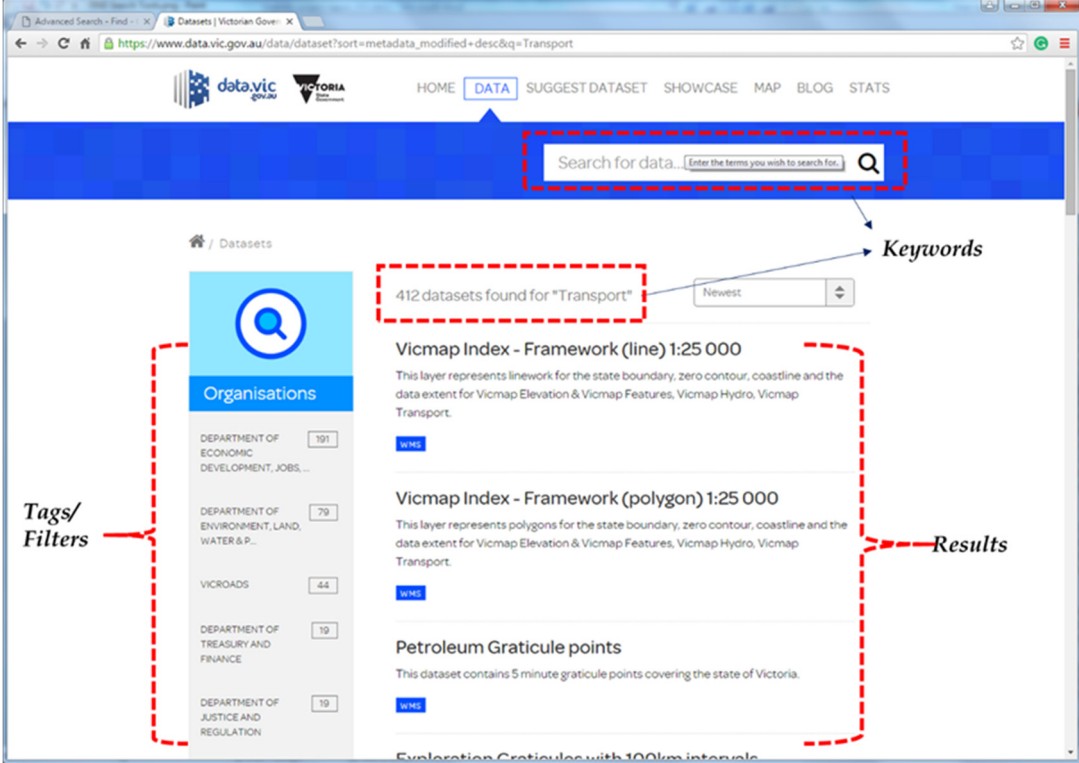

**Figure 3.** Problems identified with the Victoria Data Directory (VDD) systems in spatial data discovery.

Similar types of concludes are found in the results. From the lists the identified problems can be grouped into two types; user interface problems and metadata records problems. Table 6 summarises the problems that can be addressed to increase the usability of spatial metadata for spatial data discovery.

**Table 6.** Identified usability problems in spatial data discovery.

| User Interface Problems | Metadata Records Problems |
|---|---|
| ✓ Missing important filter for data search e.g., scale is not provided (in VDD and FIND) | ✓ Inconsistent information presented in titles e.g., scale, unknown abbreviations |
| ✓ Tags are not necessarily relevant to the data (VDD) | ✓ Keywords are different from the data description |
| ✓ Inconsistency of the results when users change the newest to most relevant (FIND) | ✓ Inconsistent abstracts e.g., contain superfluous information or too short and uninformative |
| ✓ Irrelevant results with the submitted keywords (VDD and FIND) | |
| ✓ Advanced search is not available (VDD) | |

### 4.4. User Satisfactions and Expectations

User satisfaction is not a usability attribute that can be easily evaluated, since it is related to emotions, feelings and experiences of participants during the task completions. However, it could be detected by carefully observing the data (phrases) in conjunction with researcher's notes while observing the experiments. In the process of identifying the user satisfaction, user expectations on both the spatial metadata records and user interface are observed simultaneously, since the satisfaction is determined by the gap between the expectations and the reality. Following are some of the phrases that can be used to detect user satisfaction and expectations during the spatial data discovery.

"I think, I cannot find the scale, probably I need to open one by one to see what the scale is"

"People should define some basic needs, and they have to face with this kind of search"

Similar phrases were recurring during the spatial data discovery process that can be used to detect user satisfaction as well as identify the expectations. From the above phrases, an unsatisfactory result can be detected as the participant could not find the filter scale to narrow down the results. They expected that the searching tool would provide them with a set of basic criteria to narrow down their search.

The researcher's notes also indicate that participants were not enthusiastic when they opened the VDD website, since it only provides a simple search by submitting keywords. They did not notice the tags provided on the navigation window at the left–hand side of the web page. Even when they found out about the tags, they were not happy because when they tried to use them, the results were more irrelevant than they expected. Different reactions were given by the participants when they opened the FIND website. Participants were enthusiastic when they found the advanced search in the website. However, they started to complain about the website when they received results, especially because they could not find the way to narrow down their search using the scale. Table 7 summarises user expectations for spatial data discovery.

**Table 7.** Identified user expectations for spatial data discovery.

| User Interface | Metadata Records |
|---|---|
| ✓ Simple (keywords) and advanced search based on basic criteria for spatial data, including the scale | ✓ Consistent and informative titles e.g., containing the scale and other basic information |
| | ✓ Consistent and informative abstract e.g., detailed information about data contents presented in brief textual descriptions |
| ✓ Consistent and relevant tags for the filters | |
| ✓ Relevant and consistent results presentation e.g., titles and abstracts | ✓ Consistent and relevant keywords with titles |

**5. Analysing the Outcomes of Spatial Data Selection Tasks**

Spatial data selection is a process where participants try to understand the characteristics of spatial data by reading and interpreting information presented in spatial metadata to decide whether the spatial data is suitable or fit for the given scenario or not. In their attempts to read and interpret the information, participants use their knowledge and experiences which vary between them.

The effectiveness, efficiency, usability problems, user satisfaction and expectation for spatial data selection can be identified from the protocol analysis results.

*5.1. Effectiveness*

Conclude phrases related to this process were observed to identify the result of participants' attempts on data selection.

"So, it is not very easy to make a quick decision"

"After they give you a sample then maybe you have to contact them and ask them what this data is about"

"Currently from this metadata, you are not sure about data"

"I am not sure of this. I think, I can say that I am in the middle"

"I am confident, but I need more time to read, probably I need more experience"

"You need to spend more time, and you will be more confident"

"I think it is enough for me to decide whether this is what I need or not"

"I think, if can have a look at the data itself, I might be confident"

Again, similar phrases recurred during the experiments as thoughts from participants. As in spatial discovery, it is not easy to clearly say that spatial metadata is effective or ineffective for spatial data selection, as participants gave positive, negative and neutral concludes. Nonetheless, the phrases allow us to sense a degree of ineffectiveness. For example, a participant's verbalisations vacillated during the process. At some points, the participant would be confident in their decisions, but later, lack conviction when they had a different experience with the metadata.

Based on authors' notes during the process, at first, participants looked at the metadata page as a whole to find information that might be critical to their decisions, including abstract, production date, update, producer and accuracy, and then felt confident that they could make a decision. However, after they had a detailed look at each element, they started to have doubts. For example, when they looked at the data accuracy, they were very critical and demanded detailed information. They complained about the unclear values of the accuracy, e.g., 1–10 m, and asked for an accuracy evaluation process. Participants had similar experiences with metadata from both websites, VDD and FIND. Moreover, all participants required additional information that was not provided in the metadata, including spatial data sample (access to the data), information regarding producer's reputation and reports on spatial data quality tests.

*5.2. Efficiency*

Similar to spatial discovery, participants were only able to read and to evaluate half or less of the metadata from the given tasks. They spent much of their time trying to find critical information within the metadata to assess the suitability of the data, for example, accuracy, to figure out whether the information was useful or not. They spent more time trying to interpret information that was not always relevant to their knowledge. The following phrases are examples of what participants had to deal with and what cost them time locating and interpreting information in the metadata.

"the description is not so clear and too much for data source"

"just have a list maybe of who are the providers"

"why put this in data source? class should not be in data sources"

"I think they mess (mixed up) the information (in this part of metadata)"

Accordingly, efficiency of metadata needs to be increased. Moreover, participants need more time to acquire additional information from different sources, including producers and data experts, to assist them to make clear decisions on the data.

### 5.3. Usability Problems

Data analysis results provide us with various problems faced by participants during the spatial data selection process. These problems prevent them from making a clear decision about the suitability of the data for the given scenario, as well as hampering participants in making selections in the given amount of time. Missing important content, as in maintenance date or last update, discouraging presentation of data quality and abstracts, were also issues. However, the main problem, which all participants identified, was the irrelevance and inconsistency of information returned in abstracts and in accuracy. Figure 4 illustrates the problem found in an abstract.

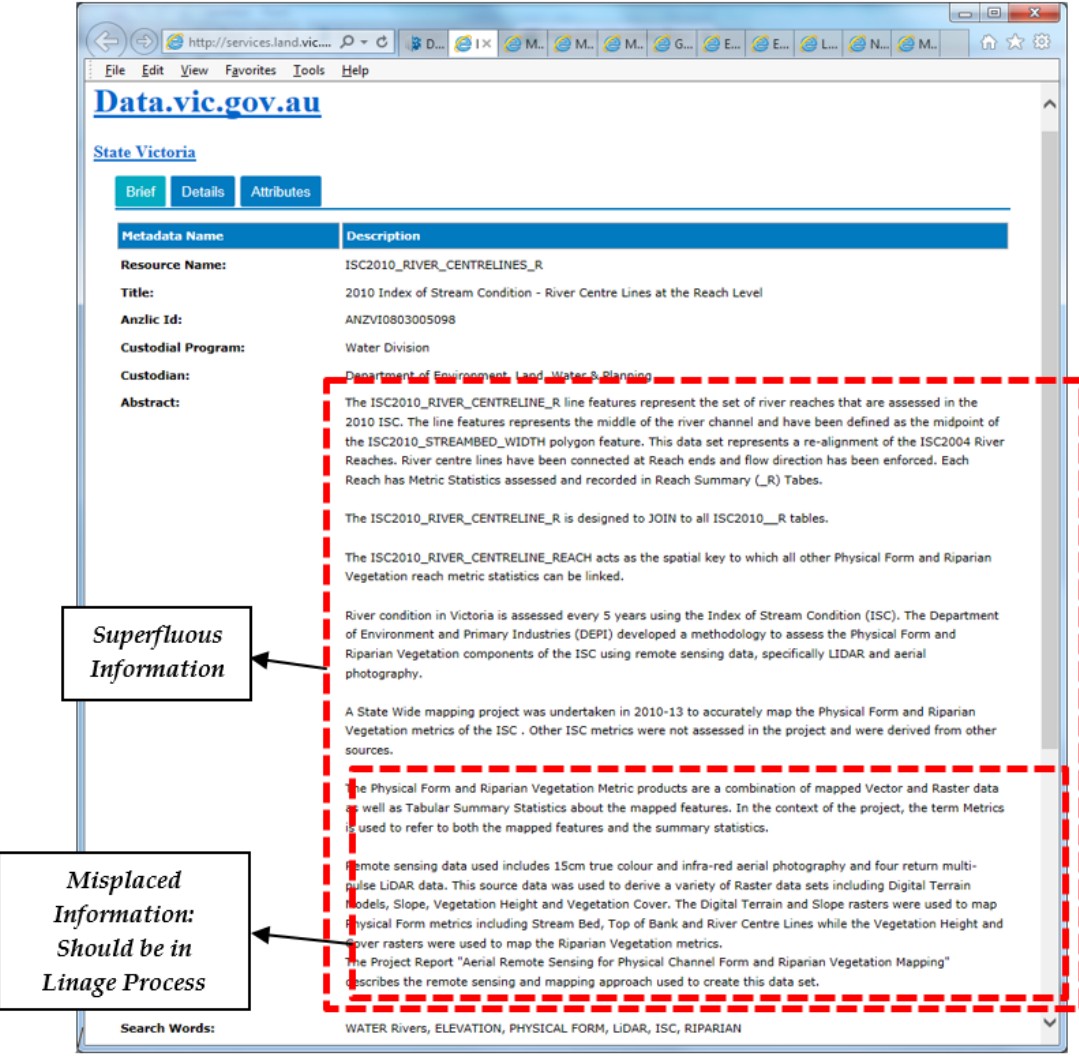

**Figure 4.** Superfluous information in an abstract.

Participants agreed that abstract plays a crucial role in spatial data selection, as they would start the process and reasoning with it. A brief, clear, easy to understand and relevant abstract, with the terminologies they know, is essential to help them understand what the data is about and how the data would fit their purposes. A good quality abstract, according to users' perceptions, would encourage and guide them to find other information about the data within the metadata. Conversely, a messy abstract would confuse and discourage them from getting more information from the metadata. Table 8 summarises the usability problems identified from the experiment.

**Table 8.** Identified usability problems in spatial data selection.

| User Interface | Metadata Records |
|---|---|
| ✓ Discouraging presentation e.g., plain textual presentation | ✓ Missing metadata contents |
| | ✓ Inconsistent content between metadata records e.g., in abstract |
| | ✓ Unfamiliar terminologies e.g., in abstract and accuracy |
| | ✓ Inconsistent length of content e.g., in abstract and accuracy |
| | ✓ Lack of information in contents e.g., no validation test for accuracy |

Besides the above listed usability problems, the data provides us with a meta–problem, that is, a problem beyond a usability problem found by participants in metadata records and user interfaces. This meta–problem was detected from the phrases below.

"I think it's the provider, because we need to know to judge, whether we can trust the data or not"

"to clearly describe the data, but I think the best way is to have a sample"

"or they give you a sample then after that maybe you have to contact them"

"if you can go and search one sample, not too much but to see what kind of data they have to be sure"

"you can read and say –definitely I need this– but when you, download or get down, you figure out that this is not my data"

The problem was related to confidence, where participants felt that information presented in metadata would only be useful to examine the information in metadata. It is not that they did not trust the metadata, but they thought that they could not base their decisions on the suitability of the spatial data solely on the information presented in the metadata. In order to make decisions, they required additional information, referred to as meta–information, including the reputation of the producers, their experiences with the data or the metadata system, that could only be obtained by live or direct interaction with the data. Similar reasoning might be found in the car sales business, where customers would look at and read the brochure or advertisements to identify potential cars that would fit their criteria. However, deciding to buy or not to buy the car is a different story and requires different information that cannot be provided by the brochures or advertisements, that is, a live experience with the car itself. Hence, test driving. The needs and expectations of this meta–information are explained in the next section.

*5.4. User Satisfactions and Expectations*

During the spatial data selection process, participants identify specific information in metadata records that could be useful to assess the suitability of the data for the given project. In doing so, participants start with reading the abstracts to get a brief understanding about the data before they continue the process. However, participants found that most of the abstracts were problematic and

discouraged them from reading and spending more time to understand the information. This led them to go to other metadata records based on their existing knowledge to identify the important information that might be useful for them to assess the suitability. Most of them went for accuracy information and found that information presented was not necessarily clear and comprehensible. Participants went for, amongst other things, information such as production date (the date created) and the boundary (geographic coverage). During the process, they kept verbalising their thoughts, which were mainly negative, about the metadata, as they found that metadata did not meet their expectations, as summarised in Table 9.

**Table 9.** Identified users' expectations for spatial data selection.

| User Interface | Metadata Records |
|---|---|
| ✓ Alternative presentation style e.g., statistical or graphical presentation for data accuracy<br><br>✓ Dynamic/intuitive presentation/pages | ✓ Brief and consistent abstracts with critical information for spatial data selection<br>✓ Use of familiar terminology to users' knowledge and experiences<br>✓ Access to data sample or preview for data self–assessment<br>✓ Access to data producers for additional information about the data |

## 6. Analysing Interviews

After completing the given tasks, participants were given a set of questions regarding the spatial data discovery and selection. The authors used these questions to explore participants' opinions and experiences about working with spatial metadata. They were encouraged to recall any critical incidents during the process and give their explanations about the incidents.

Question 1: To what extent do you agree that you can find the required spatial data?

Participants gave various answers to this question as they had different experiences and problems. As can be seen in Table 10, for both spatial data discovery systems, participants' opinions vary, with no strong indications that they successfully located the required spatial data.

**Table 10.** Participants' opinions about the results of the spatial data discovery.

| Participants' Responses | VDD | | FIND | |
|---|---|---|---|---|
| | n | % | n | % |
| Strongly agree | 0 | 0% | 0 | 0% |
| Agree | 2 | 40% | 1 | 20% |
| Either agree or disagree | 2 | 40% | 1 | 20% |
| Disagree | 0 | 0% | 1 | 20% |
| Strongly disagree | 1 | 20% | 2 | 40% |

Participants mentioned the lack of criteria as filters to narrow down their search from the earliest stage. They recalled how irritating it was when they had to look at so many listed titles and to read and check the titles one by one only to discover that the titles were inconsistently presented. When they were reminded about tags in the VDD website that could be used to filter the data, most of them were still not happy, as the tags were not easily recognised and did not provide the expected criteria, such as map scale and region (geographic coverage). They did not want to spend a lot of time searching for data that was not there. Hence, their need for complete and well–labelled filters for searching criteria to get immediate results or no results.

Question 2: To what extent do you agree that the spatial metadata (and the user interface) meet your expectations?

As can be seen in Table 11, participants' responses to the questions were mostly negative, as were their answers to the previous question. They found serious problem during the discovery process, e.g., the inconsistency between the keywords they submitted with the titles presented in the result. A participant who had experience with the discovery websites mentioned about how to improve the system, e.g., provide tags from previous users to help him identify appropriate data as they might have different terminology for the same data and share this with other users.

**Table 11.** Participants' opinions about their expectations in spatial data discovery.

| Participants' Responses | VDD | | FIND | |
|---|---|---|---|---|
| | n | % | n | % |
| Strongly agree | 0 | 0% | 0 | 0% |
| Agree | 0 | 0% | 1 | 20% |
| Either agree or disagree | 1 | 20% | 0 | 0% |
| Disagree | 3 | 60% | 2 | 40% |
| Strongly disagree | 1 | 20% | 2 | 40% |

Another participant mentioned putting more effort into making the metadata records complete, clear and consistent, both within a record and between metadata records. They recalled a specific experience where they had encountered a title with abbreviations in it that has no explanation whatsoever in the abstract nor in the rest of the records, and they had to open the metadata because the title appeared on top of the most relevant result.

Question 3: To what extent do you agree that you can determine spatial data fitness for use to decide whether you will use the data or not?

Participants with a spatial information background tended to leave the metadata page once they had information about the contents/features, the age of the data (last update) and the geographic coverage (region covered by the data). They would look for opportunities to download and get the data and perform their own assessment on the actual data or data sample, to find out whether the data was fit for their purpose or not. Or, they preferred to contact the data producer directly to get the information they required to make decisions about the data, instead of reading the metadata and basing their decisions on the metadata. Those from non–spatial backgrounds, like the civil engineer, would prefer to look at the data producer and get information regarding the reputation of the producers to be sure about the data. The participants' explanations, as can be seen in Table 12, were therefore mostly negative.

**Table 12.** Participants' opinions about the spatial metadata for suitability assessment.

| Participants' Responses | VDD | | FIND | |
|---|---|---|---|---|
| | n | % | n | % |
| Strongly agree | 0 | 0% | 0 | 0% |
| Agree | 1 | 25% | 0 | 0% |
| Either agree or disagree | 3 | 50% | 1 | 25% |
| Disagree | 1 | 25% | 4 | 75% |
| Strongly disagree | 0 | 0% | 0 | 0% |

Another participant, who was very confident with his spatial knowledge, preferred to read the report on spatial data processing, to find out information about the methodology for data collection and standards for quality validations.

Question 4: To what extent do you agree that the spatial metadata met your expectations for spatial data suitability assessment?

As can be seen from the participants' responses in Table 13, the metadata sufficiency for assessing suitability of spatial data for a certain purpose is subject to their experience with the data and their professional backgrounds. Most of the participants gave negative responses, as the metadata barely meet their expectations. Instead, they required additional information that was not part of the metadata. The next question is, to what extent the metadata could be improved to meet their expectations? Participants' responses were unclear, as they could not articulate what they expected from the metadata. Some of them, again, asked for access to sample data. Some asked for previous usages by previous users such as in reports or reviews. Some participants said that they would not decide the suitability of the data based on the metadata. They would only use it to discover the data availability and they would try to access or obtain and check the data suitability by working with the data directly.

**Table 13.** Participants' opinions about the spatial metadata related to their requirements for assessing suitability of spatial data.

| Participants' Responses | VDD | | FIND | |
|---|---|---|---|---|
| | n | % | n | % |
| Strongly agree | 0 | 0% | 0 | 0% |
| Agree | 1 | 20% | 1 | 20% |
| Either agree or disagree | 2 | 40% | 0 | 0% |
| Disagree | 2 | 40% | 3 | 60% |
| Strongly disagree | 0 | 0% | 1 | 20% |

Question 5: To what extent do you agree that you are confident to use spatial metadata again in the future for spatial data discovery and selection?

This was the last question given to participants. As Table 14 shows, the responses are, again, mostly negative. However, their explanations were not so negative. Most of the participants would still use and rely on the metadata, and data discovery systems, for searching potential spatial metadata for their future purposes. They would like to see the metadata and user interface improved by addressing identified problems and their requirements from this experiment. They still thought that metadata was important, but improvement should be made to make metadata more usable and useful to them. One participant would like to see metadata presented in different styles and presentations based on users' expertise levels, general, intermediate and experts, using different terminologies in accordance with the level of expertise. Users could choose and use the discovery system and presentation suitable for them.

**Table 14.** Participants' confidence level using spatial metadata in the future.

| Participants' Responses | VDD | | FIND | |
|---|---|---|---|---|
| | n | % | n | % |
| Strongly agree | 0 | 0% | 0 | 0% |
| Agree | 1 | 20% | 1 | 20% |
| Either agree or disagree | 2 | 40% | 0 | 0% |
| Disagree | 2 | 40% | 2 | 40% |
| Strongly disagree | 0 | 0% | 2 | 40% |

## 7. Discussion

The results and findings of the usability evaluation of established spatial metadata reveals that the metadata and the user interface remain problematic for spatial data users to discover and select spatial data. Problems such as inconsistent information in most elements in the metadata records hampered spatial data users from effectively and efficiently discovering and selecting spatial data. Another problem that can be highlighted is the irrelevance of the information presented in the metadata to the users' knowledge. The problems found in the user interface add another difficulty to the

discovery and selection process. Lack of searching criteria, irrelevant results from the submitted search, and discouraging textual presentation that is either too short or too long, make the discovery and selection process harder.

The fact that these metadata records were created following the same metadata standard ISO 19115 did not prevent the problems from happening. Most of the problems come from the free–text type of elements in spatial metadata standards such as title, abstract, keywords, data quality and data provenance. The information mainly comes from the metadata authors' understanding about the data. This indicates that the standard of capability remains lacking for maintaining the consistency of information presented in the metadata and the relevance of the information to the spatial data users' knowledge.

Another finding in spatial data selection reveals that spatial data users require more than just information from metadata to be able to select the data for their purposes. Access to actual spatial data or a sample is important for them so they can assess the suitability of the data themselves. They also think that experiences from other users who worked with the data are useful for the selection process. This is another indicator that the standards of the main guideline for spatial metadata creation should be improved.

The latest version of the ISO standard for geospatial metadata is the ISO 19115–1:2014, Geographic information—Metadata, Part 1: Fundamentals. According to the standard, a full metadata record is an aggregate of 12 metadata classes. There is information that can be of help for users that the standard does not currently have elements for accommodating; for example, user data rating and the number of downloads. The future research should be designed to address the problems and user expectations of spatial metadata to improve the usability of spatial metadata and recommend how spatial metadata standard ISO 19115 can be extended to provide a solution for better metadata usability.

## 8. Conclusions

This paper outlined results of a spatial metadata usability evaluation experiment. The results indicate that the spatial metadata is neither effective nor efficient for spatial data discovery and selection, and improvements are required of both the metadata records and the user interface. The experiment also revealed some usability problems in both metadata records and user interfaces (websites). Inconsistency and irrelevant information in the metadata records were found in the title, keywords, abstracts, data quality and other elements of the metadata. Lack of searching criteria, irrelevant results from the submitted search, and discouraging presentation of the metadata are some prominent problems found in the user interface. Those problems hampered the spatial data users from effectively and efficiently discovering and selecting spatial data. Moreover, the results also revealed that the information presented in metadata records does not suffice the needs of spatial data users to make a selection of spatial data. Previous user reviews and experiences and access to actual data or sample are also required by the users to make their selection.

Accordingly, spatial data users were not satisfied and expected that the websites would provide them with better searching tools and better results in terms of consistency of relevance of information regarding spatial data.

The results and findings from this evaluation will be used as the basis for usability improvements and for garnering spatial data users' needs and expectations from the wider community, as a future direction.

## 9. Patents

This section is not mandatory, but may be added if there are patents resulting from the work reported in this manuscript.

**Author Contributions:** Conceptualisation, Mohsen Kalantari and Syahrudin Syahrudin; Methodology, Syahrudin Syahrudin; Software, Hardi Subagyo; Validation, Syahrudin Syahrudin, Mohsen Kalantari and Abbas Rajabifard; Formal Analysis, Syahrudin Syahrudin; Investigation, Syahrudin Syahrudin; Resources, Mohsen Kalantari; Data Curation, Syahrudin Syahrudin; Writing—Original Draft Preparation, Syahrudin Syahrudin; Writing—Review and Editing, Mohsen Kalantari; Visualisation, Mohsen Kalantari; Supervision, Mohsen Kalantari, Abbas Rajabifard; Project Administration, H.H.; Funding Acquisition, Mohsen Kalantari, Abbas Rajabifard. All authors have read and agreed to the published version of the manuscript.

**Funding:** This research was funded by Australian Awards Scholarships and Australian Research Council grant number DP170100153.

**Acknowledgments:** The authors would also like to thank all the members of the Centre for Spatial Data Infrastructures and Land Administration (CSDILA) and the Centre for Disaster Management, of the University of Melbourne, for all the discussions and enjoyments.

**Conflicts of Interest:** The funders had no role in the design of the study; in the collection, analyses, or interpretation of data; in the writing of the manuscript, or in the decision to publish the results.

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
