# Peer review of "Spatial Metadata Usability Evaluation"

_ijgi, doi:10.3390/ijgi9070463_

Round 1

Reviewer 1 Report

Review of the manuscript Spatial metadata usability evaluation

The paper overall is interesting and it refers to a topic, which is both timely and original. It is well - written and of good quality. The authors clearly state the purpose of the article. The article fits the scope of IJGI Journal and will be of interest to its readers.

The literature review in Introduction section should be presented more detailed.

In line 310 there is “Figure 4.2 illustrated some of the problems identified during spatial data discovery as mentioned by the participants (…)”. There should be: “Figure 3 illustrated some of the problems identified during spatial data discovery as mentioned by the participants (…)”.

I recommend publication after minor revision.

Author Response

Thank you for reviewing the paper and for providing feedback.

As requested we added more details about the literature by specifying their findings.

We have also changed Table 4.2 to Table 3.

Reviewer 2 Report

This usability study of spatial metadata offers useful results to help in the improvement of spatial search and discovery.  It address both the discovery system design (website/repository) and the metadata elements and values (content).  Seeing the test respondents results is compelling and should encourage concrete action in improving the metadata provision of spatial metadata.

Consider adding link to the ISO standard or sharing more details in section 6 where the there is discussion test participants misunderstanding labeled elements to compare system label to standard label and definition.

line 37 - ..."for spatial data users [ADD "is"] for discovering..

line 53 "structure[d]" interviews - ADD "d"

line 56-60 - is this necessary to include?

line 86 "spatial metadata usability [ADD "is"] evaluated

line 107 "two data directory [edit to "directories"] or portals"

line 117 "Another important [delete" pice" technique]

line 117 "this study was [ADD "the use of"] audio recording...

line 147 "discovery, [add comma] and selection...

line 176 "the interview[s delete "s"] plural] data was....

line 214 "via the user-interface [- delete hyphen]...

line 397 "thoughts [switch by to "from"] participants....

line 408 ...and asked for [ADD "am"] accuracy evaluation....

line 479 Most of them [ change went to looked for] accuracy...

line 480 information presented [delete "in the accuracy:] was not necessarily..

line 482 change (the "age of spatial data") to date created...

line 508 Hence, their need for complete [ ADD and well-labeled] filters...

line 601 data or [Add "a"] sample...

Author Response

Thank you for your thorough review of the paper. We edited the paper and addressed the writing errors. As requested, we also added more details about ISO and its usability issues to Section 7.